# Incorporating Prior Knowledge into Neural Networks through an Implicit Composite Kernel

## Abstract

It is challenging to guide neural network (NN) learning with prior knowledge. In contrast, many known properties, such as spatial smoothness or seasonality, are straightforward to model by choosing an appropriate kernel in a Gaussian process (GP). Many deep learning applications could be enhanced by modeling such known properties. For example, convolutional neural networks (CNNs) are frequently used in remote sensing, which is subject to strong seasonal effects. We propose to blend the strengths of deep learning and the clear modeling capabilities of GPs by using a composite kernel that combines a kernel implicitly defined by a neural network with a second kernel function chosen to model known properties (e.g., seasonality). Then, we approximate the resultant GP by combining a deep network and an efficient mapping based on the Nyström approximation, which we call Implicit Composite Kernel (ICK). ICK is flexible and can be used to include prior information in neural networks in many applications. We demonstrate the strength of our framework by showing its superior performance and flexibility on both synthetic and real-world data sets. The code is available at: `https://anonymous.4open.science/r/ICK_NNGP-17C5/`.

## 1 Introduction

In complex regression tasks, input data often contains *multiple sources of information*. These sources can be presented in both high-dimensional (e.g. images, audios, texts, etc.) and low-dimensional (e.g. timestamps, spatial locations, etc.) forms. A common approach to learn from high-dimensional information is to use neural networks (NNs) [21, 33], as NNs are powerful enough to capture the relationship between complex high-dimensional data and target variables of interest. In many areas, NNs are standard practice, such as the dominance of Convolutional Neural Networks (CNNs) for image analysis [26, 61, 62]. In contrast, for low-dimensional information, we usually have some prior knowledge on how the information relates to the predictions. As a concrete example, consider a remote sensing problem where we predict ground measurements from satellite imagery with associated timestamps. *A priori*, we expect the ground measurements to vary periodically with respect to time between summer and winter due to seasonal effects. We would typically use a CNN to capture the complex relationship between the imagery and the ground measurements. In this case, we want to guide the learning of the CNN with our prior knowledge about the seasonality. This is challenging because knowledge represented in NNs pertains mainly to correlation between network units instead of quantifiable statements [36].

Conversely, Gaussian processes (GPs) have been used historically to incorporate relevant prior beliefs by specifying the appropriate form of its kernel function (or covariance function) [2, 54]. One approach to modeling multiple sources of information is to assign a relevant kernel function to each source of information respectively and combine them through addition or multiplication, resulting in a

Submitted to 36th Conference on Neural Information Processing Systems (NeurIPS 2022). Do not distribute.

*composite kernel function* [14]. This formulation means that specifying a kernel to match prior beliefs on one source of information is straightforward. Such composite kernel learning techniques are extensively used in many application areas such as multi-media data [40], neuroimaging [60], spatial data analysis, and environmental data analysis [28, 44]. In view of the clear modeling capabilities of GP, it is desirable to examine how a NN could be imbued with the same modeling ease.

In recent years, researchers have come up with a variety of methods to incorporate prior knowledge into NNs. These efforts can be broken into many categories, such as those that add prior information through loss terms like physics-informed NNs [32, 41]. Here, we focus on the major category of those methods that build integrated models of NNs and GPs with various structures [50, 57, 58]. Related to our proposed methodology, Pearce et al. [43] exploited the fact that a Bayesian neural network (BNN) approximates a GP to construct additive and multiplicative kernels, but they were limited to specific predefined kernels. Matsubara et al. [38] then resolved this limitation by constructing priors of BNN parameters based on the ridgelet transform and its dual, but they did not explicitly show how their approach works for data with multiple sources of information. To our knowledge, none of these existing approaches allows a modeler to choose any appropriate kernel of known prior information from multiple sources. We address this limitation by presenting a simple yet novel Implicit Composite Kernel (ICK) framework, which processes high-dimensional information using a kernel implicitly defined by a neural network and low-dimensional information using a chosen kernel function. The low-dimensional kernels are mapped into the neural network framework to create a straightforward and simple-to-learn implementation. Our key results and contributions are:

- We analytically show our ICK framework, under reasonable assumptions, is approximately equivalent to a Gaussian process regression (GPR) model with a composite kernel *a priori*.
- We demonstrate that our ICK framework yields better performance on both prediction and forecasting tasks even with very limited data.
- We compare to joint deep learning models, such as a neural network-random forest joint model, to show that ICK can flexibly capture the patterns of the low-dimensional information without deliberately designing a pre-processing procedure or complex NN structures.

Based on these contributions, we believe ICK will be useful in the context of learning from complex *hybrid* data with prior knowledge, especially in the field of remote sensing and spatial statistics.

## 2   Related Work

**Equivalence between NNs and GPs**   The equivalence between GPs and randomly initialized single-layer NNs with infinite width was first shown by Neal [42]. With the development of modern deep learning, researchers further extended this relationship to deep networks [34, 39] and convolutional neural networks (CNNs) [17]. This relationship is crucial for proving the resemblance between GPR and our ICK framework, which will be discussed in Section 4.1.

**NNs with prior knowledge**   As mentioned before, one approach to equip NNs with prior knowledge is to modify the loss function. For example, Lagaris et al. [32] solved differential equations (DEs) using NNs by setting the loss to be a function whose derivative satisfies the DE conditions. Another approach is to build integrated models of NNs and GPs. For example, Wilson et al. [58] implemented a regression network with GP priors over latent variables and made inference by approximating the posterior using Variational Bayes or sampling from the posterior using Gibbs sampling scheme. Garnelo et al. [16] introduced a class of neural latent variable models called Neural Processes (NPs) which are capable of learning efficiently from the data and adapting rapidly to new observations. Zhu et al. [10] proposed NeuralEF which can accurately approximately kernel functions by using a series of objective functions parameterized by NNs under the principle of eigen-decomposition.

**GP with composite kernels**   Composite kernel GPs are widely used in both machine learning [14, 54] and geostatistical modeling [9, 18]. GPR in geostatistical modeling is also known as *kriging* [27, 31], which serves as a surrogate model to replace expensive function evaluations. The inputs for a composite GP are usually low-dimensional (e.g. spatial distance) as GPs do not scale well with the number of samples for high-dimensional inputs [4, 5]. To overcome this issue, Pearce et al. [43] and Matsubara et al. [38] developed BNN analogue for composite GPs. Similar to these studies, our ICK framework can also be viewed as a simulation for composite GPs.

**Approximation methods for GP** For large data sets, approximation methods are needed as exact kernel learning and inference scales $\mathcal{O}(N^3)$. Nyström low-rank matrix approximation [12, 53] and Random Fourier Features [45, 46] are two of the most commonly used methods. A common technique is to choose inducing points as pseudo-inputs to efficiently approximate the full kernel matrix [49, 23]. Our work is inspired by these approximation techniques and we use them as *transformation functions* to map the kernel matrix into latent space representations in Section 4.2.

# 3 Background

Before elaborating on the details of our ICK framework, we introduce our notation, briefly go over the concepts of composite GPs, and describe the relationship between GPs and NNs.

## 3.1 Problem Setup

To formalize the problem, we have a training data set which contains $N$ data points $\boldsymbol{X} = [\boldsymbol{x}_i]_{i=1}^N = [\boldsymbol{x}_1, \boldsymbol{x}_2, ..., \boldsymbol{x}_N]^T$ and the corresponding labels of these data points are $\boldsymbol{y} = [y_i]_{i=1}^N = [y_1, y_2, ..., y_N]^T$ where $y_i \in \mathbb{R}$. Each data point $\boldsymbol{x}_i = \left\{ x_i^{(1)}, x_i^{(2)}, ..., x_i^{(M)} \right\}$ is composed of information from $M$ different sources where the $m^{th}$ source of information of the $i^{th}$ data point is denoted as $x_i^{(m)} \in \mathbb{R}^{D_m}$. Our goal is to learn a function $\hat{y}_i = f(\boldsymbol{x}_i) : \left\{ \mathbb{R}^{D_1}, \mathbb{R}^{D_2}, ..., \mathbb{R}^{D_M} \right\} \to \mathbb{R}$ which takes in a data point $\boldsymbol{x}_i$ and outputs a predicted value $\hat{y}_i$.

## 3.2 Composite GPs

A Gaussian process (GP) describes a distribution over functions [54]. A key property of GP is that it can be completely defined by a mean function $\mu(\boldsymbol{x})$ and a kernel function $K(\boldsymbol{x}, \boldsymbol{x}')$. The mean function $\mu(\boldsymbol{x})$ is often assumed to be zero for simplicity. In that case, the outcome function is

$$f(\boldsymbol{x}) \sim \mathcal{GP}\left(0, K(\boldsymbol{x}, \boldsymbol{x}')\right). \tag{1}$$

Any finite subset of these random variables has a multivariate Gaussian distribution with mean $\boldsymbol{0}$ and kernel matrix $\boldsymbol{K}$ whose entries can be calculated as $\boldsymbol{K}_{ij} = K(\boldsymbol{x}_i, \boldsymbol{x}_j)$. In many situations, the full kernel function is built by a composite kernel by combining simple kernels through addition $K_{\text{comp}}(\boldsymbol{x}, \boldsymbol{x}') = K_1(\boldsymbol{x}, \boldsymbol{x}') + K_2(\boldsymbol{x}, \boldsymbol{x}')$ or multiplication $K_{\text{comp}}(\boldsymbol{x}, \boldsymbol{x}') = K_1(\boldsymbol{x}, \boldsymbol{x}')K_2(\boldsymbol{x}, \boldsymbol{x}')$ [14]. A useful property that ICK exploits is that $K_1$ and $K_2$ can take different subparts of $\boldsymbol{x}$ as their inputs. For example, $K_{\text{comp}}(\boldsymbol{x}, \boldsymbol{x}') = K_1\left(x^{(1)}, x^{(1)'}\right) + K_2\left(x^{(2)}, x^{(2)'}\right)$ or $K_{\text{comp}}(\boldsymbol{x}, \boldsymbol{x}') = K_1\left(x^{(1)}, x^{(1)'}\right) K_2\left(x^{(2)}, x^{(2)'}\right)$. Other methods such as functional mapping are also valid if the resulting kernel matrix $\boldsymbol{K}$ is positive semidefinite (PSD) for all possible choices of data set $\boldsymbol{X}$ [47].

## 3.3 Correspondence between GPs and NNs

Neal [42] proved that a single-hidden layer network with infinite width is *exactly equivalent* to a GP over data indices $i = 1, 2, ..., N$ under the assumption that the weight and bias parameters of the hidden layer are i.i.d. Gaussian with zero mean. Lee et al. [34] and Garriga et al. [17] then extended this statement to deep neural networks and convolutional neural networks (CNNs), respectively. However, if the width (or the number of channels) of a network is finite, then these results state that the network *approximately* converges to a GP with zero mean as in the following lemma.

**Lemma 1** *Let $\boldsymbol{z} = f_{NN}\left(x^{(1)}\right) : \mathbb{R}^{D_1} \to \mathbb{R}^p$ be the latent representation extracted from $x^{(1)}$ where $p$ is the dimension of the extracted representation and $f_{NN}$ is a neural network with finite width and zero-mean i.i.d. parameters. The $k^{th}$ entry of this representation will* approximately *follow a NN-implied GP*

$$z_k = f_{NN}\left(x^{(1)}\right)_k \sim \mathcal{GP}_{approx}\left(0, K_{NN}\left(x^{(1)}, x^{(1)'}\right)\right). \tag{2}$$

That is to say, the $k^{th}$ component $z_k$ of the representation extracted by the network has zero mean $\mathbb{E}_{p(\theta^{(1)})}\left[z_{ik}^{(1)}\right] = 0$ for all $i = 1, 2, ..., N$ where $\theta$ represents the network parameters. The covariance between $z_{ik}^{(1)}$ and $z_{jk}^{(1)}$ for *different* data indices $i, j = 1, 2, ..., N$ can be approximated as

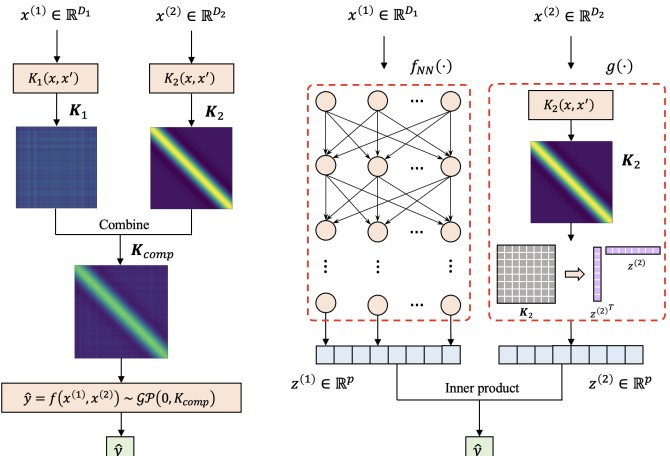

Figure 1: Given data containing 2 sources of information $x^{(1)}$ and $x^{(2)}$, we can process the data using either (**Left**) a composite Gaussian process regression (GPR) model or (**Right**) our ICK framework where $x^{(1)}$ is processed with a neural network $f_{\text{NN}}(\cdot)$ and $x^{(2)}$ is processed with $g(\cdot)$ where $g(\cdot)$ consists of a kernel function $K_2$ and some transformation which maps the kernel matrix $K_2$ into the latent space.

cov $\left( z_{ik}^{(1)}, z_{jk}^{(1)} \right) = \mathbb{E}_{p\left(\theta^{(1)}\right)} \left[ z_{ik}^{(1)} z_{jk}^{(1)} \right] \approx K_{\text{NN}} \left( x_i^{(1)}, x_j^{(1)} \right)$ where $x_i^{(1)}$ and $x_j^{(1)}$ are the corresponding inputs for the network and $K_{\text{NN}}$ is the kernel function implied by the network.

## 4 Implicit Composite Kernel (ICK) Framework

We show the structure of a composite GPR model and our ICK framework in Figure 1. To make the illustration clear, we limit ourselves to data with information from 2 different sources $\boldsymbol{x} = \left\{ x^{(1)}, x^{(2)} \right\}$ where $x^{(1)}$ is high-dimensional and $x^{(2)}$ is low-dimensional (i.e. $D_1 \gg D_2$) with some known relationship with the target $y$. We are inspired by composite GPR, which computes 2 different kernel matrices $K_1$ and $K_2$ and then combines them into a single composite kernel matrix $K_{\text{comp}}$. However, as discussed before, it is more suitable to use a NN to learn from the high dimensional information $x^{(1)}$. In our ICK framework, we process $x^{(1)}$ with a NN $f_{\text{NN}}(\cdot) : \mathbb{R}^{D_1} \to \mathbb{R}^p$ and $x^{(2)}$ with a mapping $g(\cdot) : \mathbb{R}^{D_2} \to \mathbb{R}^p$ which consists of a kernel function $K_2$ followed by a kernel-to-latent-space transformation (described in Section 4.2), resulting in two latent representations $\boldsymbol{z}^{(1)}, \boldsymbol{z}^{(2)} \in \mathbb{R}^p$. Then, we make a prediction $\hat{y}$ by doing an *inner product* between these two representations $\hat{y} = f_{\text{NN}} \left( x^{(1)} \right) \cdot g \left( x^{(2)} \right)$. Finally, the parameters of both the NN and the kernel function are learned via gradient-based optimization methods [3, 30].

In the sections below, we first analytically show that our ICK framework is *approximately* equivalent to a composite GPR model *a priori* using a multiplicative kernel between the kernel implicitly defined by the NN on $x^{(1)}$ and the chosen kernel on $x^{(2)}$. This theory is used to motivate the model form. The model will deviate from the GP solution after learning, but we note that recent work suggests that the predictions from a trained NN may not vary too much from its GP equivalent [34]. We then show how we implement the kernel-to-latent-space transformation in detail. Here, we note that we apply ICK for multiplicative kernels but note that an additive kernel may be constructed using the methods of Pearce et al. [43].

### 4.1 Resemblance between Composite GPR and ICK

We will analytically prove the following theorem for data with information from 2 different sources $\boldsymbol{x} = \left\{ x^{(1)}, x^{(2)} \right\}$ for clarity, and we note this theorem can be straightforwardly extended to $M > 2$.

**Theorem 1** *Let $f_{NN} : \mathbb{R}^{D_1} \to \mathbb{R}^p$ be a NN function with random weights and $g : \mathbb{R}^{D_2} \to \mathbb{R}^p$ be a mapping function, and define an inner product $\hat{y}$ between the representations $\boldsymbol{z}^{(1)} = f_{NN} \left( x^{(1)} \right)$ and $\boldsymbol{z}^{(2)} = g \left( x^{(2)} \right)$. Then this inner product approximately follows a composite GPR model*

$$\hat{y} = f_{ICK} \left( x^{(1)}, x^{(2)} \right) = f_{NN} \left( x^{(1)} \right) \cdot g \left( x^{(2)} \right) \sim \mathcal{GP}_{approx} \left( 0, K_{NN}^{(1)} K^{(2)} \right), \qquad (3)$$

*if $g$ includes the following deterministic kernel-to-latent-space transformation*

$$K^{(2)} \left( x_i^{(2)}, x_j^{(2)} \right) \approx \boldsymbol{z}_i^{(2)^T} \boldsymbol{z}_j^{(2)} = g \left( x_i^{(2)} \right)^T g \left( x_j^{(2)} \right), \qquad (4)$$

where $K_{NN}^{(1)}$ is a NN-implied kernel and $K^{(2)}$ is any valid kernel of our choice.

To prove Theorem 1, we first make the following assumption.

**Assumption 1** *For latent representations $\boldsymbol{z}_i^{(m)}$ and $\boldsymbol{z}_j^{(m)}$ extracted from different data points $\boldsymbol{x}_i$ and $\boldsymbol{x}_j$ where $i \neq j$ and $m \in \{1, 2\}$, the interactions between different entries of $\boldsymbol{z}_i^{(m)}$ and $\boldsymbol{z}_j^{(m)}$ can be reasonably ignored. In other words, let $\theta^{(m)}$ be the parameters of the network or the kernel function which takes in $x^{(m)}$ and outputs $\boldsymbol{z}^{(m)}$, we have $\mathbb{E}_{p\left(\theta^{(m)}\right)}\left[z_{ik}^{(m)} z_{jl}^{(m)}\right] = 0$ for all $k \neq l$.*

With Assumption 1 and Lemma 1, let $\Theta = \left\{\theta^{(1)}, \theta^{(2)}\right\}$ represent the parameters of the ICK framework, we can calculate the covariance between $\hat{y}_i$ and $\hat{y}_j$ for different data indices $i \neq j$ as follows

$$\text{cov}(\hat{y}_i, \hat{y}_j) = \mathbb{E}_{p(\Theta)}[\hat{y}_i \hat{y}_j] - \mathbb{E}_{p(\Theta)}[\hat{y}_i]\mathbb{E}_{p(\Theta)}[\hat{y}_j] \tag{5}$$

$$= \mathbb{E}_{p(\Theta)}\left[\left(\sum_{k=1}^{p} z_{ik}^{(1)} z_{ik}^{(2)}\right)\left(\sum_{k=1}^{p} z_{jk}^{(1)} z_{jk}^{(2)}\right)\right] \tag{6}$$

$$= \mathbb{E}_{p(\Theta)}\left[\sum_{k=1}^{p} \sum_{l=1}^{p} z_{ik}^{(1)} z_{jl}^{(1)} z_{ik}^{(2)} z_{jl}^{(2)}\right] \tag{7}$$

$$= \mathbb{E}_{p(\Theta)}\left[\sum_{k=1}^{p} z_{ik}^{(1)} z_{jk}^{(1)} z_{ik}^{(2)} z_{jk}^{(2)}\right] \tag{8}$$

$$= \sum_{k=1}^{p} \mathbb{E}_{p\left(\theta^{(1)}\right)}\left[z_{ik}^{(1)} z_{jk}^{(1)}\right] \mathbb{E}_{p\left(\theta^{(2)}\right)}\left[z_{ik}^{(2)} z_{jk}^{(2)}\right] \tag{9}$$

$$\approx K_{\text{NN}}^{(1)}\left(x_i^{(1)}, x_j^{(1)}\right) \sum_{k=1}^{p} \mathbb{E}_{p\left(\theta^{(2)}\right)}\left[z_{ik}^{(2)} z_{jk}^{(2)}\right]. \tag{10}$$

Here, from Equation 5 to Equation 6, we use the statement $\mathbb{E}_{p\left(\theta^{(1)}\right)}\left[z_{ik}^{(1)}\right] = 0$ from Lemma 1 and the independence between $\theta^{(1)}$ and $\theta^{(2)}$, which leads to $\mathbb{E}_{p(\Theta)}[\hat{y}_i] = \mathbb{E}_{p(\Theta)}[\hat{y}_j] = 0$. From Equation 7 to Equation 8, we get rid of all the cross terms under Assumption 1. From Equation 8 to Equation 9, we again make use of the independence between $\theta^{(1)}$ and $\theta^{(2)}$. From Equation 9 to Equation 10, we use the statement $\mathbb{E}_{p\left(\theta^{(1)}\right)}\left[z_{ik}^{(1)} z_{jk}^{(1)}\right] \approx K_{\text{NN}}\left(x_i^{(1)}, x_j^{(1)}\right)$ from Lemma 1. If the kernel-to-latent-space transformation in $g(\cdot)$ is *deterministic*, we can remove the expectation sign from the summation term in Equation 10 and the covariance can be further expressed as

$$\text{cov}(\hat{y}_i, \hat{y}_j) \approx K_{\text{NN}}^{(1)}\left(x_i^{(1)}, x_j^{(1)}\right)\left(\boldsymbol{z}_i^{(2)^T} \boldsymbol{z}_j^{(2)}\right) = K_{\text{NN}}^{(1)}\left(x_i^{(1)}, x_j^{(1)}\right) K^{(2)}\left(x_i^{(2)}, x_j^{(2)}\right), \tag{11}$$

which means that $\hat{y}$ approximately follows a GP with composite kernel $K_{\text{comp}}(\boldsymbol{x}_i, \boldsymbol{x}_j) = K_{\text{NN}}^{(1)}\left(x_i^{(1)}, x_j^{(1)}\right) K^{(2)}\left(x_i^{(2)}, x_j^{(2)}\right)$ *a priori*. This completes our proof of Theorem 1.

## 4.2 Kernel-to-latent-space Transformation

We now show how we can construct an appropriate mapping $g(\cdot)$ that approximately satisfies the assumed form of (4) and is used in the derivation of ICK from (10) to (11). Here we adopt two methods, Nyström approximation and Random Fourier Features (RFF), to map the kernel matrix into the latent space. Below, we give the formulations and results for the Nyström method, and give the methods and results for RFF in Appendix B. According to Yang et al. [59], the Nyström method will yield much better performance than RFF if there exists a large gap in the eigen-spectrum of the kernel matrix. In our applications, we also observe a large eigen-gap (see details in Appendix C) and Nyström method does generalize much better than RFF. We name our framework with Nyström method and *r*andom Fourier Features ICK*y* and ICK*r*, respectively.

### 4.2.1 Nyström Method

The main idea of Nyström method [53] is to approximate the kernel matrix $\boldsymbol{K} \in \mathbb{R}^{N \times N}$ with a much smaller low-rank matrix $\boldsymbol{K}_q \in \mathbb{R}^{q \times q}$ where $q \ll N$ so both the computational and space complexity of kernel learning can be significantly reduced

$$\boldsymbol{K} \approx \hat{\boldsymbol{K}} = \boldsymbol{K}_{nq} \boldsymbol{K}_q^{-1} \boldsymbol{K}_{nq}^T. \tag{12}$$

The entries of $\boldsymbol{K}_q$ and $\boldsymbol{K}_{nq}$ can be calculated as $(\boldsymbol{K}_q)_{ij} = K(\hat{x}_i, \hat{x}_j), i, j \in \{1, 2, ..., q\}$ and $(\boldsymbol{K}_{nq})_{ij} = K(x_i, \hat{x}_j), i \in \{1, 2, ..., N\}, j \in \{1, 2, ..., q\}$, respectively. $x$ represents the original data points and $\hat{x}$ represents pre-defined inducing points (or pseudo-inputs [49]). In our study, these inducing points are chosen by defining an evenly spaced vector over the range of original data points. By performing Cholesky decomposition $\boldsymbol{K}_q^{-1} = \boldsymbol{U}^T\boldsymbol{U}$, where $\boldsymbol{U} \in \mathbb{R}^{q \times q}$, $\hat{\boldsymbol{K}}$ is

$$\hat{\boldsymbol{K}} = \boldsymbol{K}_{nq}\boldsymbol{K}_q^{-1}\boldsymbol{K}_{nq}^T = \boldsymbol{K}_{nq}\boldsymbol{U}^T\boldsymbol{U}\boldsymbol{K}_{nq}^T = \left(\boldsymbol{U}\boldsymbol{K}_{nq}^T\right)^T\left(\boldsymbol{U}\boldsymbol{K}_{nq}^T\right). \tag{13}$$

Therefore, if we set the number of inducing points to be $q = p$, then we can use $\boldsymbol{z}_i \triangleq \boldsymbol{U}\left(\boldsymbol{K}_{np}^T\right)_{:,i}$ as a kernel-to-latent-space transformation because each element in $\boldsymbol{K}$ approximately satisfies (4) as stated in Theorem 1: $K(x_i, x_j) = K_{ij} \approx \hat{K}_{ij} = \boldsymbol{z}_i^T\boldsymbol{z}_j$. Conveniently, modern deep learning frameworks can propagate gradients through the Cholesky operation, making it straightforward to update the kernel parameters with gradient methods. Note that as we increase the number of inducing points $p$, the approximation error between $\boldsymbol{K}$ and $\hat{\boldsymbol{K}}$ decreases. However, it is not recommended to set $p$ very large as updating the Cholesky decomposition requires $\mathcal{O}(p^3)$. The empirical impact of $p$ on computational time and performance is shown in Appendix E. In our experiments, only mild values of $p$ are necessary and the impact on computational is relatively small.

## 5 Experimental Results

We evaluate ICK$y$ on 3 different data sets: a synthetic data set, a remote sensing data set, and a data set obtained from UCI Machine Learning Repository [13]. Note that in all the 3 experiments, our ICK$y$ framework only consists of 2 kernels (i.e. $M = 2$), one NN-implied kernel and one chosen kernel function with trainable parameters. To verify that ICK$y$ can work with more than 2 kernels, we create another synthetic data set with 3 kernels and show the corresponding results in Appendix A. In addition, the experimental results for ICK$r$ is provided in Appendix B. All experiments are conducted on a computer cluster equipped with a GeForce RTX 2080 Ti GPU. The implementation details of all the experiments in this section are provided in Appendix G.

### 5.1 Synthetic Data

To verify that ICK$y$ can simulate a *multiplicative kernel*, we create a synthetic data set $y \sim \mathcal{GP}(0, K_1K_2)$ containing 3000 data points where $x^{(1)} \in [0, 1]$ is the input for the linear kernel $K_1$ and $x^{(2)} \in [0, 2]$ is the input for the *spectral mixture* kernel [56] $K_2$ with 2 components. With ICK$y$, we process $x^{(1)}$ with a single-hidden-layer NN and $x^{(2)}$ with a spectral mixture kernel function. We evaluate ICK$y$ on both a *prediction* task (where we first randomly shuffle the data points and do a 50:50 train-test split) and a *forecasting* task (where we use only the data points with $x^{(2)} < 0.6$ for training and the rest for testing).

We then compare ICK$y$ with two models: a plain multi-layer perceptron (MLP) applied to the concatenated features and a novel multi-layer perceptron-random forest (MLP-RF) joint model employed by Zheng et al. [61] where MLP learns from $x^{(1)}$ and RF learns from $x^{(2)}$. We believe MLP-RF serves as a good benchmark model as it is a joint model with similar architecture to our ICK$y$ framework. To see how ICK$y$ simulates the spectral mixture kernel, we plot only $x^{(2)}$ against the predicted value of $y$ as shown in Figure 2. As can be seen from the figure, in the prediction task (top row), plain MLP only captures the linear trend. MLP-RF only captures the mean of the spectral mixture components. In contrast, our ICK$y$ framework captures both the mean and the variance of the spectral mixture kernel. In the forecasting task (bottom row), ICK$y$ also outperforms plain MLP and MLP-RF as it approximately captures the rising trend in the range of $x^{(2)} \in [0.6, 1]$. When $x^{(2)} > 1$, ICK$y$ is unable to confidently extrapolate, so it starts to "*fail gracefully*," by which we mean that the prediction reverts to the mean of the prior (e.g., no observed information case.), just as would be expected in a GP. However, we do not evaluate the posterior distribution to get a full sense of the posterior uncertainty.

We also test plain MLP, MLP-RF, and ICK$y$ on the prediction task using different number of training samples. As displayed in Figure 3, ICK$y$ yields the smallest error among all the 3 frameworks even with very limited data. In addition, to test the robustness of ICK$y$, we conduct the same experiments on another synthetic data set in Appendix D to confirm that ICK$y$ can simulate an *additive kernel*.

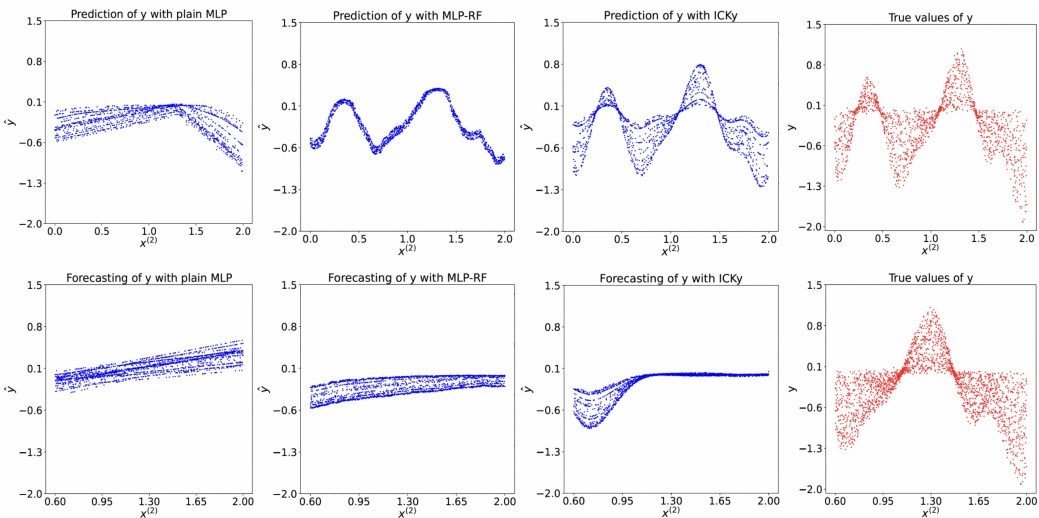

Figure 2: Prediction (top row) and forecasting (bottom row) of $y \sim \mathcal{GP}(0, K_1 K_2)$, where $x^{(1)}$ is input to a linear kernel $K_1$ and $x^{(2)}$ is input to a spectral mixture kernel $K_2$. We plot $x^{(2)}$ against the predicted $y$. We show results from a plain MLP (left column), MLP-RF (middle left column), and ICK$y$ framework (middle right column), and we compare to the true values of $y$ (right column).

## 5.2 Remote Sensing Data

We believe ICK$y$ will be particularly useful for remote sensing applications. In this experiment, we collect remote sensing data from 51 air quality monitoring (AQM) stations located in the National Capital Territory (NCT) of Delhi and its satellite cities over the period from January 1, 2018 to June 30, 2020 (see Appendix F for notes on data availability). Each data point $\boldsymbol{x} = \{x, t\}$ contains 2 sources of information: a three-band natural color (red-blue-green) satellite image $x$ as the high-dimensional information and the corresponding timestamp as the low-dimensional information. Note that we convert the timestamps into numerical values $t$ (where the day 2018-01-01 corresponds to $t = 0$) before feeding them into the models. Our goal is to predict the ground-level PM$_{2.5}$ concentration $\hat{y} = f(x, t)$ using both sources of information.

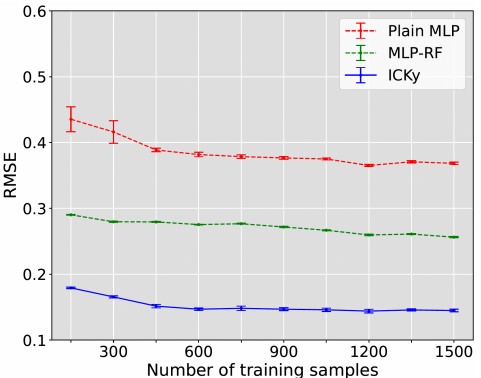

Figure 3: Prediction error of plain MLP, MLP-RF, and ICK$y$ with different amount of training data generated by $y = \mathcal{GP}(0, K_1 K_2)$.

We split the train and test data set based on $t$. Specifically, we use all the data points with $t \geq 500$ for testing and the rest for training. As PM$_{2.5}$ varies with time on a yearly basis, we use an *exponential-sine-squared* kernel with a period of $T = 365$ (days) to process the low-dimensional information $t$. The satellite images are processed with a CNN. Figure 4 shows the true versus the forecasted PM$_{2.5}$ values by both ICK$y$ and 2 benchmarks: a Convolutional Neural Network-Random Forest (CNN-RF) joint model [61, 62] (similar to the MLP-RF model in Section 5.1, where RF learns the temporal variation of PM$_{2.5}$ and CNN captures the spatial variation of PM$_{2.5}$ from satellite images) and a *carefully designed* CNN-RF model that maps $t$ into two new features, $\sin(2\pi t/365)$ and $\cos(2\pi t/365)$, to explicitly model seasonality. As can be seen, ICK$y$ outperforms both benchmarks with the highest correlation coefficients and the lowest errors on the forecasting task. Specifically, regular CNN-RF joint model fails to forecast PM$_{2.5}$ as shown in Figure 4a. After including seasonality, CNN-RF performs significantly better as shown in Figure 4b, but the forecasted PM$_{2.5}$ values are still less smooth than those from ICK$y$ (Figure 4c) due to the discontinuous nature of the RF regressor [6, 19]. We also visualize these results in the form of time series in Appendix F

Table 1: Correlation and error statistics of ICK$y$ and other joint deep models with both convolutional and attention-based architectures on the PM$_{2.5}$ forecasting task

|  | Spearman R ↑ | Pearson R ↑ | RMSE ↓ | MAE ↓ |
|---|---|---|---|---|
| CNN-RF [61] | 0.48 | 0.32 | 70.09 | 54.28 |
| ViT-RF [11] | 0.42 | 0.32 | 70.58 | 55.15 |
| Seasonal CNN-RF | 0.65 | 0.73 | 52.60 | 39.25 |
| Seasonal ViT-RF | 0.66 | 0.74 | 49.92 | 36.22 |
| Seasonal DeepViT-RF [63] | 0.68 | 0.76 | 48.87 | 35.32 |
| Seasonal MAE-ViT-RF [22] | 0.68 | 0.76 | 48.43 | 34.92 |
| CNN-ICK$y$ | **0.70** | **0.77** | **47.15** | **32.84** |
| ViT-ICK$y$ | 0.66 | **0.77** | 47.17 | 34.00 |
| DeepViT-ICK$y$ | 0.62 | 0.73 | 48.68 | 34.10 |

We note that the inner product operation in ICK is similar in mathematical structure to attention-based mechanisms [51] popular in many deep learning frameworks. Therefore, we compare ICK$y$ with 4 attention-based benchmarks that we constructed based off of a Vision Transformer (ViT) [11] architecture, including ViT-RF, Seasonal ViT-RF, Seasonal DeepViT-RF [63], and Seasonal MAE-ViT-RF where ViT is pre-trained by a Masked Autoencoder [22], as displayed in Table 1. These use the same RF and sinusoidal mappings as described previously to input the temporal information into the model. We note that we are unaware of Vision Transformers being used in this manner, and that all these models represent novel formulations. It can be observed that standard ViT-RF model fails to forecast PM$_{2.5}$ without seasonality incorporated, just as in CNN-RF. After introducing seasonality by mapping $t$ into sinusoidal features, ViT-based joint models yield higher correlation and smaller error than CNN-RF but still underperform CNN-based ICK$y$. We also considered using a ViT-based architecture for the CNN ICK$y$, and observed similar performance in these ICK$y$ variants.

## 5.3 UCI Machine Learning Repository Data

To see if our ICK$y$ framework generalizes to other domains, we acquire another data set containing the normalized productivity and corresponding features of garment workers from the UCI Machine Learning Repository. Imran et al. [1] employed a dense MLP with 2 hidden layers to predict the worker productivity with collected features such as date, team number, targeted productivity, etc. To test our ICK$y$ framework, we separate out the *date* and use it as the low-dimensional information. The rest of the features (excluding the temporal information) are then concatenated together to serve as the high-dimensional information. Observing that the *daily averaged* worker productivity has an approximate *monthly trend*, we again use an *exponential-sine-squared* kernel. The network architecture of ICK$y$ is the same as that of the two-hidden-layer MLP benchmark. To demonstrate the strength of ICK$y$ compared to other methods that equip NNs with GPs, we also add 2 additional benchmarks here: a Gaussian Neural Process (GNP) [7] and an Attentive Gaussian Neural Process (AGNP) [29]. Based on the results shown in Table 2, ICK$y$ has the best performance when the period parameter of the kernel is set to be $T = 30$ (days) and it outperforms both MLP and NP benchmarks by almost **one order of magnitude**. When we set $T = 2$ or $T = 7$, this improvement is less significant, which aligns with our initial observation that the daily averaged productivity has a monthly seasonal trend. It is also worth noting that the GNP benchmarks here yield larger errors than MLPs. A possible explanation is that GNP does not allow explicit assignment of a stationary kernel (as the kernel models a posterior covariance) so it is hard for GNP to identify specific patterns in the data such as seasonality without being given the pattern *a priori*.

Table 2: Prediction error of actual worker productivity on the test data set with ICK$y$ and other benchmark models (MLPs and NPs)

|  | MSE ↓ ($*10^{-3}$) | MAE ↓ ($*10^{-2}$) | MAPE ↓ |
|---|---|---|---|
| MLP [1] | 20.16 ± 1.26 | 9.93 ± 0.36 | 17.30 ± 0.82 |
| Cyclic MLP | 20.97 ± 1.98 | 10.16 ± 0.77 | 17.48 ± 1.37 |
| GNP [7, 37] | 57.25 ± 4.31 | 19.39 ± 0.94 | 29.58 ± 1.63 |
| AGNP [29] | 43.11 ± 5.95 | 14.38 ± 0.88 | 22.59 ± 1.42 |
| ICK$y$, $T = 2$ | 3.43 ± 1.42 | 4.85 ± 1.00 | 6.74 ± 1.37 |
| ICK$y$, $T = 7$ | 0.44 ± 0.13 | 1.43 ± 0.15 | 2.22 ± 0.21 |
| ICK$y$, $T = 30$ | **0.31 ± 0.09** | **1.17 ± 0.14** | **1.79 ± 0.22** |

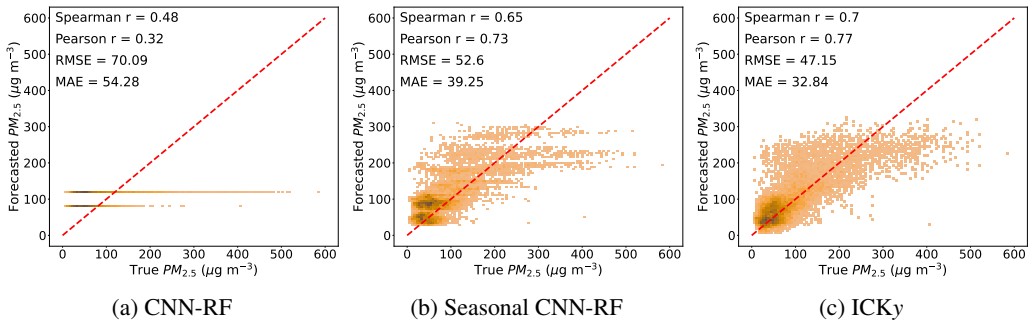

|  (a) CNN-RF | (b) Seasonal CNN-RF | (c) ICK*y* |

Figure 4: Density plots of the true PM$_{2.5}$ concentrations against the **forecasted** PM$_{2.5}$ concentrations for $t \geq 500$ using (a) a CNN-RF joint model [61, 62], (b) a CNN-RF joint model with seasonality incorporated, and (c) our ICK*y* framework

## 6 Discussion

**Efficiency, Flexibility, and Generalization** Compared to exact composite GP models which scale $\mathcal{O}(N^3)$, the training process of our ICK framework is more efficient as it leverages standard back-propagation to learn both the paramters of NN and the kernel function. In addition, the network architecture of ICK can be very simple, as can be seen in all 3 experiments of ours, which further reduces its time and space complexity. Besides efficiency, our ICK framework is more flexible compared to other joint models (i.e. BNNs and CNN-RF). To be specific, the BNNs implemented by Pearce et al. [43] cannot simulate complicated kernels such as the spectral mixture kernel we use in Section 5.1. The CNN-RF joint model implemented by Zheng et al. [61] requires us to carefully design the input pre-processing procedure. Also, ICK generalizes well to unseen data even with very limited training samples. There is a potential concern that ICKy may run into computational challenges when a large number of inducing points are required. This was not a problem in our experiments, but in large scale models this could be tackled by considering conjugate gradient methods, which have been recently popular in GP inference [15].

**Limitations** A major limitation of ICK lies in our method of combining latent representations as the nature of inner product (i.e. the effect of multiplying small numbers) may cause *vanishing gradient* problems when we have a large number of sources of information (i.e. $M$ is large). Furthermore, this paper only discusses the theoretical relationship between ICK and composite GPR *a priori*. This relationship will not exactly hold true *a posteriori*, although empirical results [34] and theoretical results [24] in slightly different contexts suggest that they may be close. Future work will evaluate this gap by exploring Bayesian neural networks and *a posteriori* properties.

**Broader Impacts** We believe our framework is extensively applicable to regression problems in many fields of study involving high-dimensional data and multiple sources of information with perceptible trends, such as remote sensing, spatial statistics, or clinical diagnosis.

## 7 Conclusion

This paper presents a novel yet surprisingly simple Implicit Composite Kernel (ICK) framework to learn from *hybrid* data containing both high-dimensional information and low-dimensional information with prior knowledge. We first analytically show the resemblance between ICK and composite GPR models and then conduct experiments using both synthetic and real-world data. It appears that ICK outperforms various benchmark models in our experiments with lowest prediction errors and highest correlations even with very limited data. Overall, we show that our ICK framework is exceptionally powerful when learning from *hybrid* data with our prior knowledge incorporated, and we hope our work can inspire more future research on joint machine learning models, enhancing their performance, efficiency, flexibility, and generalization capability.

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

Figure A1: Scatter plots of the true values of $y$ against the predicted values of $y$ using our ICK$y$ framework with (a) one source, (b) 2 sources, and (c) 3 sources of information

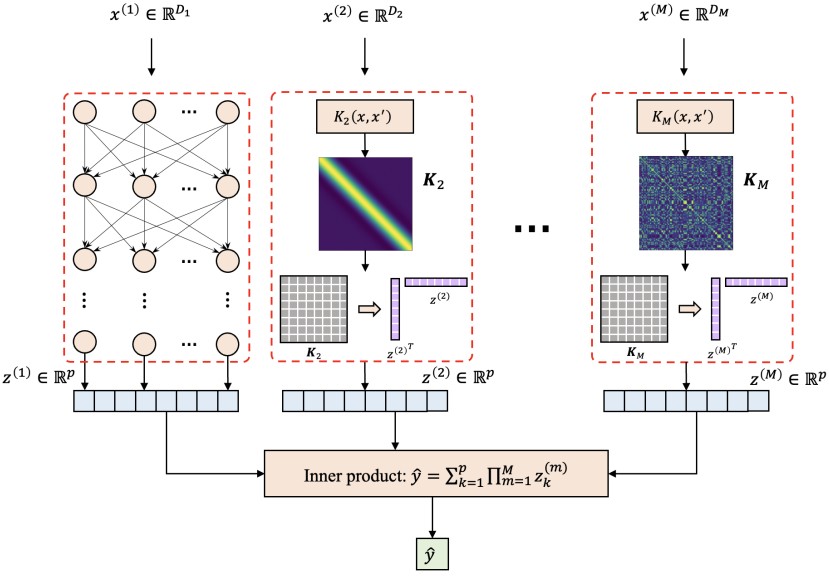

Figure A2: Given data containing $M$ sources of information $\boldsymbol{x} = \left\{x^{(1)}, x^{(2)}, ..., x^{(M)}\right\}$, we can process the data using our ICK framework where high-dimensional information (e.g. $x^{(1)}$ in the figure) is processed using a neural network and low-dimensional information (e.g. $x^{(2)}$ in the figure) is processed using a kernel function followed by Nyström or RFF transformation.

## A   ICK with More Than Two Kernels

Besides the visualization presented in Figure 1, we also show our ICK framework for processing data $\boldsymbol{x} = \left\{x^{(1)}, x^{(2)}, ..., x^{(M)}\right\}$ with $M > 2$ sources of information in Figure A2. Here $K^{(2)}, ..., K^{(M)}$ represent different types of kernels with trainable parameters. The final prediction is calculated by a chained inner product of all extracted representations $\hat{y} = \sum_{k=1}^{p} \prod_{m=1}^{M} z_k^{(m)}$.

To confirm that ICK$y$ can work with more than 2 kernels, we construct another synthetic data set containing 3000 data points in total. Each input $\boldsymbol{x} = \left\{x^{(1)}, x^{(2)}, x^{(3)}\right\}$ has 3 sources of information. The output $y$ is generated by $y = x^{(3)} \tanh\left(2x^{(1)} \cos^2\left(\pi x^{(2)}/50\right)\right) + \epsilon$ where $\epsilon$ is a Gaussian noise term. We process $x^{(1)}$ with a small single-hidden-layer NN, $x^{(2)}$ with an *exponential sine squared* kernel, and $x^{(3)}$ with a *radial-basis function* (RBF) kernel. Figure A1 shows the prediction results as we progressively add more sources of information into our ICK$y$ framework with corresponding kernel functions. It can be observed that ICK$y$ yields both smallest error and highest correlation

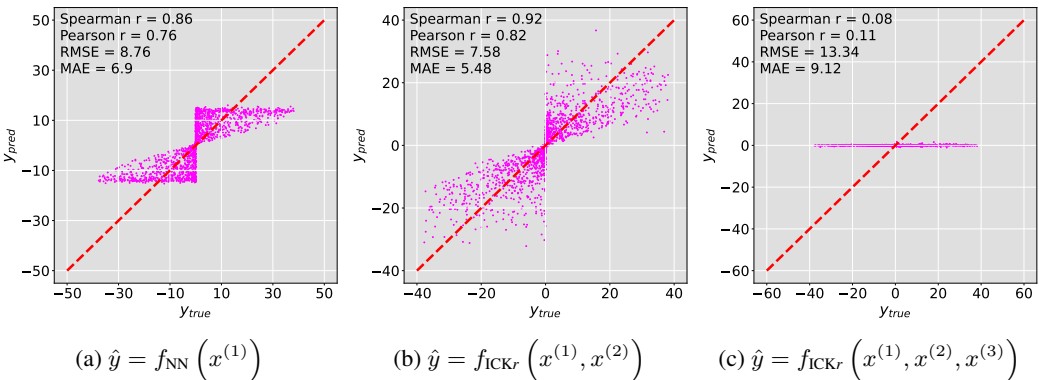

(a) $\hat{y} = f_{\text{NN}}\left(x^{(1)}\right)$    (b) $\hat{y} = f_{\text{ICK}r}\left(x^{(1)}, x^{(2)}\right)$    (c) $\hat{y} = f_{\text{ICK}r}\left(x^{(1)}, x^{(2)}, x^{(3)}\right)$

Figure B1: Scatter plots of the true values of $y$ against the predicted values of $y$ using our ICK$r$ framework with (a) one source of information, (b) 2 sources of information, and (c) 3 sources of information. Note that here we use RFF for kernel-to-latent-space transformation.

## B  Random Fourier Features

### B.1  Methodology

Random Fourier Features (RFF) is another popular approximation method used for kernel learning [45]. Unlike the Nyström method which approximates the entire kernel matrix, RFF directly approximates the kernel function $K$ using some randomized feature mapping $\phi : \mathbb{R}^{D_m} \to \mathbb{R}^{2d_m}$ such that $K\left(x_i^{(m)}, x_j^{(m)}\right) \approx \phi\left(x_i^{(m)}\right)^T \phi\left(x_j^{(m)}\right)$. To obtain the feature mapping $\phi$, based on Bochner's theorem, we first compute the Fourier transform $p$ of kernel $K$

$$p(\omega) = \frac{1}{(2\pi)^{D_m}} \int_{-\infty}^{+\infty} e^{-j\omega^T \delta} K(\delta) d\delta, \tag{14}$$

where $\delta = x_i^{(m)} - x_j^{(m)}$. Then we draw $d_m$ i.i.d. samples $\omega_1, \omega_2, ..., \omega_{d_m}$ from $p(\omega)$ and construct the feature mapping $\phi$ as follows

$$\begin{aligned}
\phi\left(x^{(m)}\right) &\equiv \\
d_m^{-1/2} &\left[\cos\left(\omega_1^T x^{(m)}\right), ..., \cos\left(\omega_{d_m}^T x^{(m)}\right), \sin\left(\omega_1^T x^{(m)}\right), ..., \sin\left(\omega_{d_m}^T x^{(m)}\right)\right].
\end{aligned} \tag{15}$$

Since $\phi\left(x^{(m)}\right) \in \mathbb{R}^{2d_m}$, we need to set $d_m = p/2$ when using RFF as a kernel-to-latent-space transformation. In addition, since RFF involves sampling from a distribution, the kernel parameters are thus not directly differentiable and we need to apply a reparameterization trick [35] to learn those parameters.

### B.2  Experimental Results

#### B.2.1  Synthetic Data

We use the same toy data set where each data point $\boldsymbol{x} = \left\{x^{(1)}, x^{(2)}, x^{(3)}\right\}$ contains 3 sources of information as described in Appendix A. Also, we use the same types of kernels as those in ICK$y$ as discussed in Appendix A. The only difference here is that we use RFF instead of Nyström method to transform the kernel matrix into the latent space in ICK$r$ framework.

The results are displayed in Figure B1. It can be observed that when we add in only the side information $x^{(2)}$ along with the *exponential sine squared* kernel, both the correlation and the predictive performance are improved (though not as good as the results from ICK$y$ as shown in Figure A1).

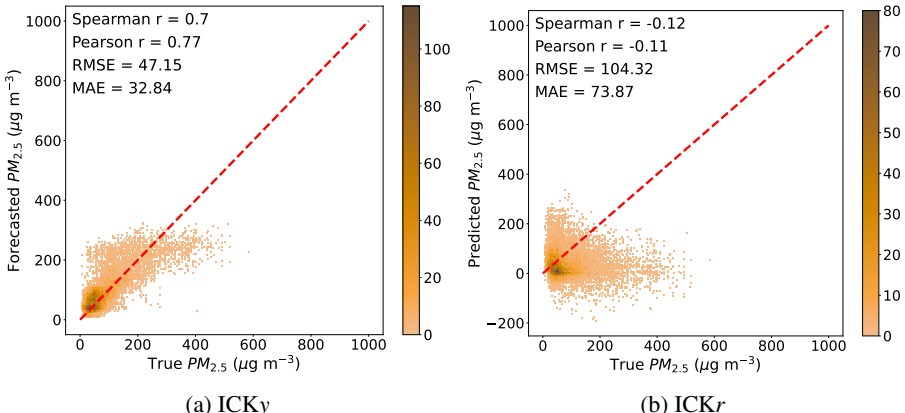

(a) ICK*y*                                    (b) ICK*r*

Figure B2: Density plots of the true PM$_{2.5}$ concentrations against the forecasted PM$_{2.5}$ concentrations for $t \geq 500$ using our ICK framework with (a) ICK*y* and (b) ICK*r*

However, after we further include $x^{(3)}$ with the *RBF* kernel, we realize that the parameters of ICK*r* become very hard to optimize and it fails to make valid predictions and starts to guess randomly around zero.

### B.2.2    Remote Sensing Data

We also try ICK*r* on the *forecasting* task using the remote sensing data (see Section 5.2) and compare the results with those from ICK*y*. Each data point $\boldsymbol{x} = \{x, t\}$ contains a satellite image $x$ as the high-dimensional information and its corresponding timestamp $t$ as the low-dimensional information. The satellite images are processed with a two-layer CNN and the timestamps are processed with an *exponential-sine-squared* kernel with a period of $T = 365$ (days). As can be observed from Figure B2, ICK*r* yields much higher error compared to ICK*y*.

## C    Estimated Kernel Matrix and its Eigen-spectrum

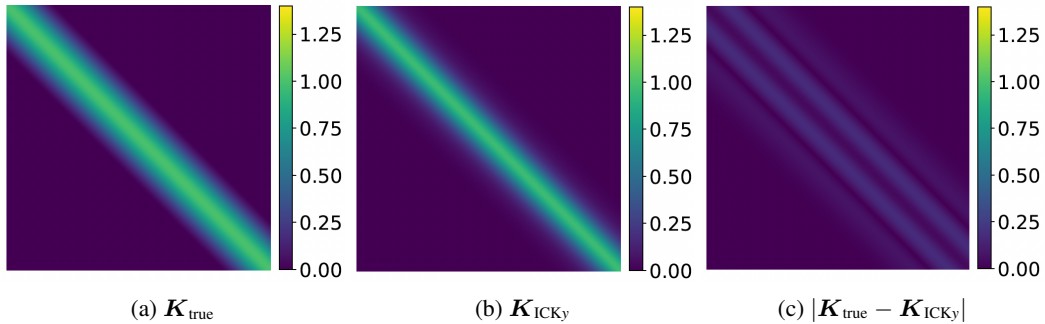

(a) $\boldsymbol{K}_{\text{true}}$                    (b) $\boldsymbol{K}_{\text{ICK}y}$                    (c) $|\boldsymbol{K}_{\text{true}} - \boldsymbol{K}_{\text{ICK}y}|$

Figure C1: Visualization of (a) True matrix (b) estimated matrix by our ICK*y* framework, and (c) absolute difference between the true and estimated matrix for the spectral mixture kernel

We first examine whether ICK*y* can retrieve the spectral mixture kernel in the prediction task. After fitting the parameters of the spectral mixture kernel in ICK*y*, we compute the kernel matrix $\boldsymbol{K}_{\text{ICK}y}$ using these learned parameters and compare it with the true kernel matrix $\boldsymbol{K}_{\text{true}}$ by calculating the absolute difference between them as displayed in Figure C1. As can be observed, $\boldsymbol{K}_{\text{ICK}y}$ and $\boldsymbol{K}_{\text{true}}$ are similar and their absolute difference is relatively small, indicating that ICK*y* can approximately retrieve the spectral mixture kernel.

Yang et al. [59] studied the fundamental difference between Nyström method and Random Fourier Features (RFF). They conclude that Nyström-method-based approaches can yield much better generalization error bound than RFF-based approaches if there exists a large gap in the eigen-spectrum of the kernel matrix. This phenomenon is mainly caused by how these two methods construct their basis functions. In particular, the basis functions used by RFF are sampled from a Gaussian distribution that is independent from the training examples, while the basis functions used by the Nyström method are sampled from the training samples so they are data-dependent. In our synthetic data experiments, we train our ICK framework using a batch size of 50. The eigenvalues of the kernel matrices computed from the first 4 batches of the synthetic data set are displayed in Figure C2. It can be observed that the first few eigenvalues of the kernel matrix are much larger than the remaining eigenvalues. Namely, there exists a large gap in the eigen-spectrum of the kernel matrix, which helps explain why ICK$y$ has a much better performance than ICK$r$.

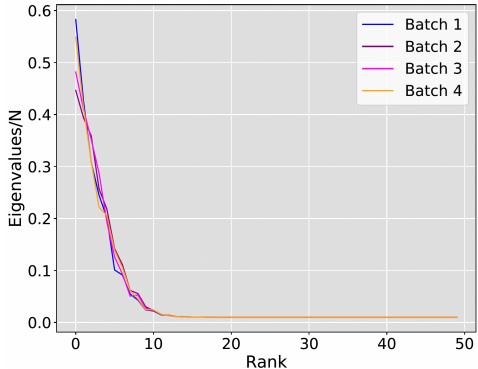

Figure C2: Eigenvalues of the kernel matrix computed from the first 4 batches of training data

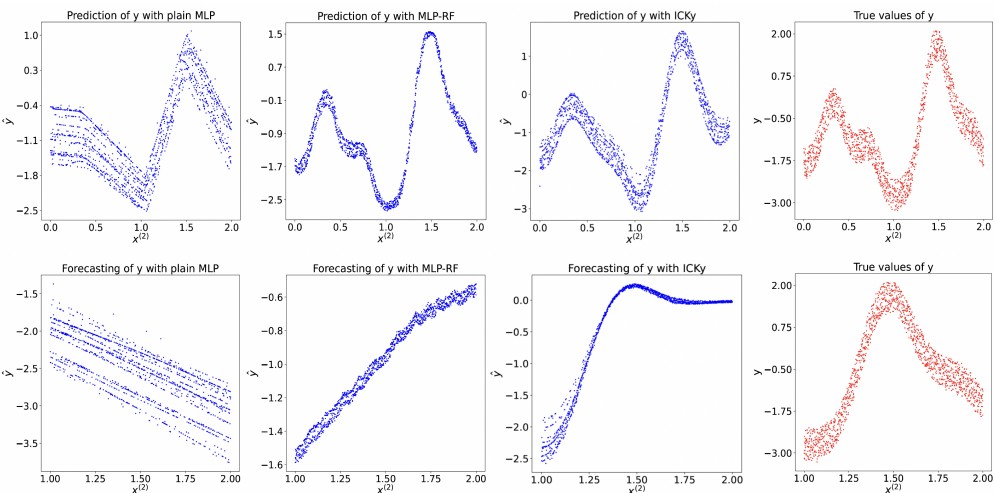

Figure D1: Prediction (top row) and forecasting (bottom row) of $y \sim \mathcal{GP}(0, K_1 + K_2)$, where $x^{(1)}$ is input to a linear kernel $K_1$ and $x^{(2)}$ is input to a spectral mixture kernel $K_2$. Here we only plot $x^{(2)}$ against the predicted $y$. We implement 3 types of models: plain MLP (left column), MLP-RF (middle left column), and our ICK$y$ framework (middle right column), and we compare the results with the true values of $y$ (right column).

## D Simulation of an Additive Kernel

While ICK is designed to capture multiplicative kernels, we evaluated how well it could capture additive kernels. We conduct experiments using another synthetic data set generated by an additive kernel $y \sim \mathcal{GP}(0, K_1 + K_2)$ with the same training settings. As shown in Figure D1, ICK$y$ again outperforms plain MLP and MLP-RF in both the prediction and the forecasting tasks. Moreover, we again test plain MLP, MLP-RF, and ICK$y$ on the prediction task using different number of training samples. As displayed in Figure D2, ICK$y$ yields the smallest error among all the 3 frameworks. Also, the performance gap between ICK$y$ and the other 2 benchmark models shrink as we feed in more training data. Therefore, we conclude ICK$y$ is robust enough to simulate both additive and multiplicative kernels.

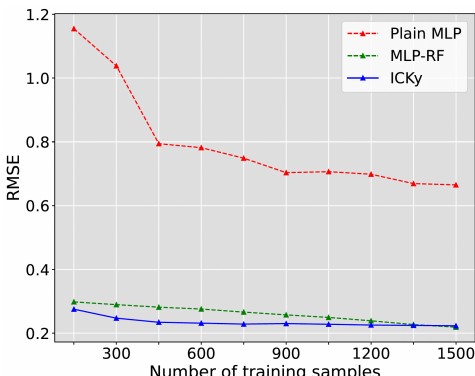

Figure D2: Prediction error of plain MLP, MLP-RF, and ICK$y$ with different amount of training data generated by $y \sim \mathcal{GP}(0, K_1 + K_2)$

## E Number of Inducing Points

As discussed in Section 4.2.1, as we increase the number of inducing points $p$, we expect the approximation error between the true kernel matrix $\boldsymbol{K}$ and the approximated kernel matrix $\hat{\boldsymbol{K}}$ to decrease. Here, we empirically show how the value of $p$ impacts our predictions. In Figure E1a, we plot the prediction error of $\hat{y} = f_{\text{ICK}y}\left(x^{(1)}, x^{(2)}, x^{(3)}\right)$ against the number of inducing points using the synthetic data generated in Appendix A. As can be observed, the prediction error drops sharply as we raise $p$ from a small value (e.g. $p = 2$). When $p$ is relatively large, increasing $p$ yields smaller improvement on the predictions. Additionally, in Figure E1b, we plot the total training time against $p$. The total training time is dependent on how long a single iteration takes and the total number of epochs required. We note that once $p > 80$ the training time is relatively flat, which is due to the fact that the total computation in the Cholesky is less than the computation in the neural network. Interestingly, it appears that when $p$ is very small, ICK$y$ takes longer to converge due to the need for many more epochs. As we increase $p$, the training time goes down and then goes up again due to the

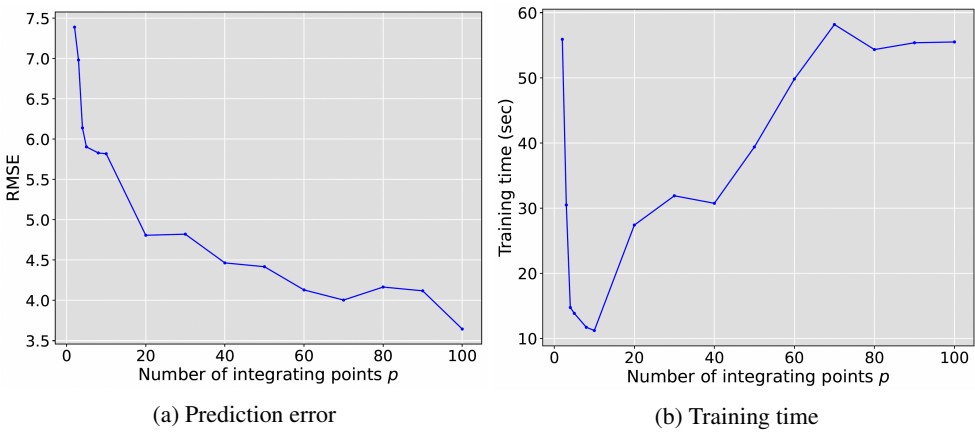

(a) Prediction error        (b) Training time

Figure E1: Plots of (a) prediction error and (b) training time of $\hat{y} = f_{\text{ICK}y}\left(x^{(1)}, x^{(2)}, x^{(3)}\right)$ against the number of inducing points $p$

computational complexity, i.e. $\mathcal{O}(p^3)$, of the Cholesky decomposition. Based on these observations, we are overly concerned about the computational complexity for reasonable values of $p$.

## F   Visualization of Remote Sensing Data as Time Series

To better illustrate the results in Section 5.2, we visualize those results in the form of time series. As shown in Figure F1, the plain CNN-RF model does not work as it tends to forecast constant $\text{PM}_{2.5}$ values. In contrast, both the seasonal CNN-RF model and our ICK$y$ framework captures the overall trend of the true daily averaged $\text{PM}_{2.5}$ values, but the forecasted values by ICK$y$ are smoother and yield smaller error.

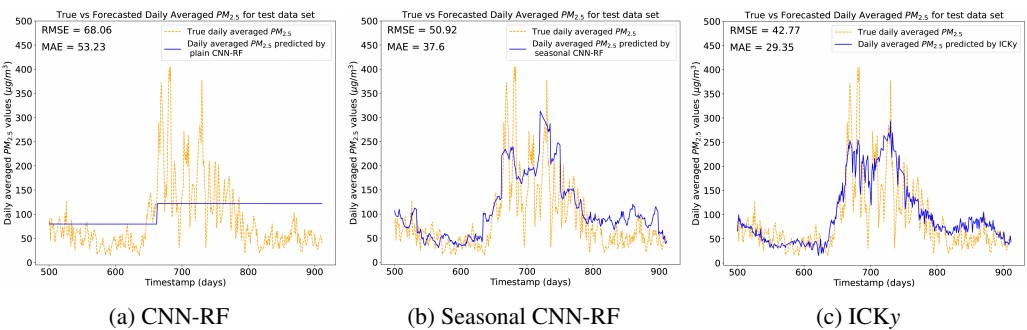

(a) CNN-RF        (b) Seasonal CNN-RF        (c) ICK$y$

Figure F1: Time series visualization of the true against the forecasted *daily averaged* $\text{PM}_{2.5}$ concentrations for $t \geq 500$ using (a) a CNN-RF joint model [61, 62], (b) a CNN-RF joint model with seasonality incorporated, and (c) our ICK$y$ framework

## G   Experimental Details

### G.1   Synthetic Data

We use the GPytorch package [15] to generate the synthetic data. Before feeding $x^{(1)}$ into MLP, we first map $x^{(1)}$ into higher dimension using an unsupervised algorithm called Totally Random Trees Embedding [48]. All the MLP structures in this experiment (including those in MLP-RF and ICK$y$) contain one single fully connected (FC) layer of width 1000, which serves as a simple benchmark since a one-hidden-layer MLP can only capture linear relationship between the input and output. For model training, we optimize a Mean Squared Error (MSE) objective using Adam optimizer [30] with a weight decay of 0.1.

Table 3: Model architecture and training details for remote sensing data experiment in Section 5.2

| | Backbone architecture details | Output FC layers dimension | Optimizer |
|---|---|---|---|
| CNN-RF | # Conv blocks = 1, # Channels = 16, Kernel size = 3, Stride = 1 | 1000, 512, 512, 1 | Adam $\beta_1 = 0.9$ $\beta_2 = 0.999$ |
| ViT-RF | # Transformer blocks = 6, # Attention heads = 8, Dropout ratio = 0.1 | 1000, 512, 512, 1 | Adam $\beta_1 = 0.9$ $\beta_2 = 0.999$ |
| DeepViT-RF | # Transformer blocks = 6, # Attention heads = 8, Dropout ratio = 0.1 | 1000, 512, 512, 1 | Adam $\beta_1 = 0.9$ $\beta_2 = 0.999$ |
| MAE-ViT-RF | # Transformer blocks = 6, # Attention heads = 8, Dropout ratio = 0.1, Masking ratio = 0.75 | 1000, 512, 512, 1 | Adam $\beta_1 = 0.9$ $\beta_2 = 0.999$ |
| CNN-ICK$y$ | # Conv blocks = 1, # Channels = 16, Kernel size = 3, Stride = 1 | 1000, 512, $p$ | SGD momentum = 0.9 |
| ViT-ICK$y$ | # Transformer blocks = 6, # Attention heads = 8, Dropout ratio = 0.1 | 1000, 512, $p$ | SGD momentum = 0.9 |
| DeepViT-ICK$y$ | # Transformer blocks = 6, # Attention heads = 8, Dropout ratio = 0.1 | 1000, 512, $p$ | SGD momentum = 0.9 |

## G.2 Remote Sensing Data

The model architecture and training details are listed in Table 3. Here $p$ denotes the length of latent representations $z$ as discussed in Section 4. Note that we use stochastic gradient descent (SGD) optimizer with a momentum of 0.9 for ICK$y$ as we realize that SGD helps ICK$y$ find a local minimum on the objective more efficiently. We use MSE objective for ICK$y$ and all benchmark models in this experiment.

## G.3 UCI Machine Learning Repository Data

The MLPs (including the MLP part in ICK$y$) in this experiment share the same structure as the one used in [1], which consist of 3 hidden layers of width 128, 32, and 32, respectively. For plain MLP, cyclic MLP, and ICK$y$, we use the mean absolute error (MAE) objective to put less weight on the outliers and thus enhance the model performance. For GNP and AGNP, we maximize a biased Monte Carlo estimate of the log-likelihood objective as discussed in [37]. All these objectives are optimized by an Adam optimizer with $\beta_1 = 0.9$ and $\beta_2 = 0.999$.

# H    Adapting ICK for Classification

While regression tasks are the primary motivation for this paper, there are many ways to adapt GPR for classification tasks. For example, a binary classification model can be created by using a sigmoid [55] or probit link [8] on the output of the GP. Succinctly, given a function $f(\boldsymbol{x}) \sim \mathcal{GP}\left(0, K(\boldsymbol{x}, \boldsymbol{x}')\right)$, the binary outcome probability is be given as $p(y = 1|f(x)) = \sigma(f(x))$. Likewise, a multiple classification model can be constructed by using a multi-output GP (or multiple GPs) and putting the outputs through a softmax function [55] or multinomial probit link [20]. This strategy can be summarized by calculating $C$ different functions $f_c(x) \sim \mathcal{GP}\left(0, K(\boldsymbol{x}, \boldsymbol{x}')\right)$ for $c = 1, ..., C$, where $C$ is the number of classes, and then calculating the class probabilities through a link function, $p(y|x) = \text{softmax}([f_1(x), f_2(x), ..., f_C(x)])$.

This same logic can be used to construct a multiple classification model from ICK$y$. Succinctly, let $r_c = f_{NN,c}(x^{(1)}) \odot z_c^{(2)}$, where $f_{NN,c}$ denotes a neural network specific to the $c^{th}$ class and $z_c^{(2)}$ represents the Nyström approximation specific to the kernel for the $c^{th}$ class. We note that often in

a multi-output case the kernel parameters are shared, and so $z_c^{(2)}$ would be an identical vector for each class. Then, the output probabilities for a data sample as $p(y|x) = \text{softmax}([r_1, \ldots, r_C])$. This framework is learned with a cross-entropy loss.

To provide *proof-of-concept* of this multiple classification strategy, we implemented this model on a version of Rotating MNIST. In this task, a dataset was created by rotating each image in the dataset by a uniform random value $\phi \in [0, 2\pi)$, thus creating a dataset with 60,000 images each with an associated rotation covariate $\phi$. We implemented the above multiple classification model with a periodic kernel over the rotation angle. This strategy yielded an accuracy of 92.3% on the validation data. This is lower than methods such as spatial transformers [25] that report accuracy greater than 99%. However, those models explicitly use the fact that the information is simply rotated, whereas ICK is modeling a smooth transformation in the prediction function as a function of angle. This ICK classification model is much closer in concept to the way Rotating MNIST is used to evaluate unsupervised domain adaptation. While the evaluation strategy is different than our random validation set, the state-of-the-art accuracy on unsupervised domain adaption is 87.1% [52]. Due to the lack of complete and fair comparisons, we are not claiming that ICK$y$ is state-of-the-art for classification, but ICK$y$'s classification model does seem reasonable and viable based upon this result.

# I   Generation, Accessibility, and Restrictions of the Data

The synthetic data $y \sim \mathcal{GP}(0, K_1 K_2)$ in Section 5.1 and $y \sim \mathcal{GP}(0, K_1 + K_2)$ in Appendix D are generated using the GPyTorch package. The remote sensing data in Section 5.2 is downloaded using PlanetScope API whose content is protected by copyright and/or other intellectual property laws. To access the data on PlanetScope, the purchase of an end-user license is required. When this manuscript is accepted, we will provide the codes we used to acquire the data. The UCI machine learning repository data we use in Section 5.3 has an open access license, meaning that the data is freely available online.

