# OpenReview forum: "Incorporating Prior Knowledge into Neural Networks through an Implicit Composite Kernel"
_NeurIPS.cc/2022/Conference — NeurIPS 2022 Submitted_

### Official Review · Reviewer_UfZP · 2022-06-13

**Rating:** 7
**Confidence:** 3
**Soundness:** 4 excellent
**Presentation:** 4 excellent
**Contribution:** 3 good

**Summary:**

The authors propose ICK, a regressor that can naturally incorporate priors over low-dimensional subsets of the input. A high-dimensional input is processed with a neural network to produce a latent vector, and the low-dimensional input produces a latent vector of the same size via a Gaussian process (GP), where the kernel function incorporates the known priors. The GP is efficiently integrated using the Nystrom method and Cholesky decomposition.

**Questions:**

As mentioned in related work, there are at least 1-2 other methods that equip NNs with priors using GPs. Can you compare with them or at least explain when/why they would be less effective than your approach?
It's not entirely clear to me what integral is being approximated with Nystrom quadrature. Is it the expectation in (10)? Might be helpful to briefly mention somewhere in 4.2.1 when you talk about integrating points.
Consider rescaling the numbers in Table 1 so there aren't so many decimal places.
Perhaps the conjugate gradient method is a feasible alternative to Cholesky decomposition if p needs to be larger? For future consideration.
Line 90 chose -> choose

**Limitations:**

Yes. Assumption 1 seems reasonable to me, but the authors should try to characterize the cases where it doesn't hold. Perhaps a rigorous condition is hard to derive, but is there any intuitive argument that the cases where it doesn't hold are ill-behaved?

**Strengths And Weaknesses:**

Strengths
The approach is an elegant combination of GPs and neural networks for incorporating useful priors, supported by a number of experiments.
The paper is well-written and the figures are very clear.
Re: vanishing gradients as a limitation. I actually think your approach improves on the naive approach wrt vanishing gradients! Normally a network may ignore low-dimensional inputs even if they are informative, but your projection of both high-dimensional and low-dimensional data sources into a common latent space encourages them to be treated more equally.

Weaknesses
One of the nice properties of GPs is that they can capture uncertainty in the prediction. However, the toy experiment suggests that this uncertainty is perhaps lost or not well calibrated. For example, I'm not sure that reverting to 0 (Fig. 3 bottom) counts as "failing gracefully" when the standard deviation is so small. Why is it confident in its bad prediction, and is there a way to mitigate this? It would be nice to see an experiment/metric that also demonstrates that your method matches distributions well.
It may be helpful to discuss what kinds of priors are amenable to being represented as the kernel function of a GP. It seems that this work focuses on continuous scalar variables where the prior represents periodicity, and the experiments are a little limited in this respect. Discuss some other types of priors and their corresponding kernels, and if possible add an experiment along this line.

---

> ### Author Response · Authors · 2022-08-02
> **Response to Reviewer UfZP**
>
> -- I'm not sure that reverting to 0 (Fig. 3 bottom) counts as "failing gracefully" when the standard deviation is so small. Why is it confident in its bad prediction, and is there a way to mitigate this?
>
> We note that this reverting to the mean behavior is idealized for the point estimate prediction and would match the prediction that you would expect from a GP.  Therefore, this is little uncertainty about the mean estimate as the network is determining that little is known in this case and is therefore confident that the prior mean is the best estimator.  However, as the reviewer points out, fully evaluating the uncertainty and the posterior distribution is necessary to fully evaluate the out-of-distribution estimates.  For now, we have added a clarifying comment in to those results in Section 5.1, and we will evaluate this property more fully in a full Bayesian implementation in future work.
>
> -- As mentioned in related work, there are at least 1-2 other methods that equip NNs with priors using GPs. Can you compare with them or at least explain when/why they would be less effective than your approach?
>
> In our revision, we add 2 more benchmarks in the revised version of our paper based on Gaussian Neural Process (GNP), which is a class of latent variable models which are capable of learning efficiently from data and adapting rapidly to new observations. However, as stated by [1], the kernel function in GNP cannot be stationary as it models a posterior covariance, which makes it more challenging to specify an appropriate kernel function. Thus, we find that straightforwardly choosing the kernel in ICK results in an easier algorithm to specify and implement. As shown in Section 5.3, GNP underperforms ICKy by a large margin as it fails to capture the seasonality in time dimension. Another related work is the Bayesian NNs developed by Pearce et al. [2] which models their GP counterparts. However, as we discussed in Discussion section, their framework cannot simulate complicated kernels such as periodic with linear decay or spectral mixture.
>
> References:
> [1] Markou, Stratis, et al. "Efficient Gaussian Neural Processes for Regression." arXiv preprint arXiv:2108.09676 (2021).
> [2] Pearce, Tim, et al. "Expressive priors in Bayesian neural networks: Kernel combinations and periodic functions." Uncertainty in artificial intelligence. PMLR, 2020.
>
> -- It's not entirely clear to me what integral is being approximated with Nystrom quadrature. Is it the expectation in (10)? Might be helpful to briefly mention somewhere in 4.2.1 when you talk about integrating points.
>
> The Nystrom quadrature approximates satisfies the quantity in Equation 4, which is then used in the sum in (10) over z^{(2)}.  We have added a clarifying comment in Section 4.2.1 to alleviate future confusion.
>
> -- Consider rescaling the numbers in Table 1 so there aren't so many decimal places. Line 90 chose -> choose
>
> We have fixed the typos and updated the format of Table 1 (Table 2 in the new version) so there are now only 2 decimal places. Thank you for your suggestion.
>
> -- Perhaps the conjugate gradient method is a feasible alternative to Cholesky decomposition if p needs to be larger? For future consideration.
>
> We agree with the reviewer that this is an interesting suggestion, and have added a comment about it in the Discussion section.
>
> -- Assumption 1 seems reasonable to me, but the authors should try to characterize the cases where it doesn't hold. Perhaps a rigorous condition is hard to derive, but is there any intuitive argument that the cases where it doesn't hold are ill-behaved?
>
> Assumption 1 is a rather mild assumption (see response to Reviewer pXCj) that holds true with most initialization procedures. We have not determined an experiment that breaks this structure in a natural way, so we are unaware of any ill-behaved cases where it doesn’t hold.

---

> > ### Comment · Reviewer_UfZP · 2022-08-04
> > **Great work**
> >
> > Thanks authors for your helpful response. This work is quite strong, and I look forward to the Bayesian variant which I think could be very high impact.

---

### Official Review · Reviewer_4n3e · 2022-07-01

**Rating:** 7
**Confidence:** 3
**Soundness:** 3 good
**Presentation:** 4 excellent
**Contribution:** 3 good

**Summary:**

The paper proposes a way to combine prior knowledge inside deep networks. It assumes data comes from different sources (at least one low-dimensional source and one high-dimensional source), and prior knowledge is known about the low-dimensional one (e.g., seasonality). They build a composite kernel where the low-dimensional data is processed with a user-defined kernel, while the high-dimensional one with a neural network. The former is then processed though kernel approximation methods to obtain a latent representation which is multiplied by the embedding of the neural network to get the final prediction. They perform several experiments ranging from a synthetic dataset to a realistic use case in remote sensing.

**Questions:**

- I think computational time should be discussed more. The Cholesky decomposition is cubic in the number of pseudo-inputs, and the overhead required by differentiating through it is unclear to me.
- "Incorporating Prior Knowledge" is a pretty strong title and motivation for the paper, since the technique is feasible only in the very specific setting described in the summary. It is impossible (as far as I understand) to include generic prior information if it is relative to the high-dimensional data (e.g., an image). I think the claims in the paper should be downsized accordingly.
- Since the kernel is included in the autodiff process, I wonder whether it is also possible to optimize some of its parameters (e.g., the seasonality) automatically. Have the authors considered this point?

**Strengths And Weaknesses:**

The idea is simple but (as far as I know, as I am not an expert in neural Gaussian Processes) novel. The paper is easy to read, the technique sound, and the experiments are interesting and relatively clear in the benefit of the method.

---

> ### Author Response · Authors · 2022-08-02
> **Response to Reviewer 4n3e**
>
> -- I think computational time should be discussed more. The Cholesky decomposition is cubic in the number of pseudo-inputs, and the overhead required by differentiating through it is unclear to me.
>
> To elaborate on the computational time, we added an additional experiment to Appendix E to show the effect that the number of pseudo-inputs has on the total computation time. The total time is dependent on the per-iteration time, which is what the Cholesky directly impacts, and the total number of epochs.  We note that the training time is not heavily dependent on the complexity of the Cholesky decomposition, which is a relatively small computation compared to the neural networks, and instead is largely dependent on the how the # of pseudo-inputs impacts the number of epochs. It appears that when # pseudo-inputs is very small, the model takes more steps, which leads to a longer training time compared to larger values.  After around 80 pseudo-inputs, the algorithm is fairly stable and the total training is fairly flat.
>
> -- "Incorporating Prior Knowledge" is a pretty strong title and motivation for the paper, since the technique is feasible only in the very specific setting described in the summary. It is impossible (as far as I understand) to include generic prior information if it is relative to the high-dimensional data (e.g., an image). I think the claims in the paper should be downsized accordingly.
>
> We concede the reviewers’ point that the title does not convey that our proposed method only covers cases whether there is a clear assumption/kernel model form on part of the data and combines it with a standard neural network. This does cover many applications in geostatistics, so it is not too narrow of a task either.  Regardless, we struggle to think of a title that would be clearer and are happy to consider a suggestion from the reviewer.
>
> -- Since the kernel is included in the autodiff process, I wonder whether it is also possible to optimize some of its parameters (e.g., the seasonality) automatically. Have the authors considered this point?
>
> Yes, in our implementation of ICK, we can freely determine whether the kernel parameters (e.g. length scale, period, noise level, etc.) are trainable or not.

---

> > ### Comment · Reviewer_4n3e · 2022-08-04
> > **Comment on title**
> >
> > "Regardless, we struggle to think of a title that would be clearer and are happy to consider a suggestion from the reviewer."
> > I agree it's not trivial, I had in mind something along "incorporating prior knowledge from auxiliary data", but I am unsure whether it's clearer. Anyway, it was not impacting my evaluation, so it's okay.

---

### Official Review · Reviewer_qDxM · 2022-07-04

**Rating:** 4
**Confidence:** 4
**Soundness:** 2 fair
**Presentation:** 2 fair
**Contribution:** 2 fair

**Summary:**

The authors introduce a neural network construction that forms predictions by computing inner products between a hidden representation generated by a neural network and a kernel-derived basis of functions.  This construction is motivated by a desire to combine the flexibility of neural networks with the domain-specific knowledge that can be encoded in kernels (e.g. seasonality). The properties of the resulting method are explored on three data sets.

**Questions:**

- For M>2 the authors suggest using a "chained inner product" to form predictions (Appendix A). The authors also state (line 295ff) that "A major limitation of ICK lies in our method of combining latent representations as the nature of inner product (i.e. the effect of multiplying small numbers) may cause vanishing gradient problems when we have a large number of sources of information (i.e. M is large)." Have you tried M>3? When do you expect ICK to break down?
- What happens when Assumption 1 (line 158ff) does not hold? Can you test this experimentally with an artificial dataset that breaks this assumption?

**Limitations:**

The authors do not discuss any potential negative societal impacts of their work.

**Strengths And Weaknesses:**

The main strength of this submission, as I see it, is that one of the empirical comparisons includes a remote sensing data set, which would seem to be a rich test bed for these kinds of methods---though it should be noted that accessing this data set requires a PlanetScope license.

Unfortunately, the weaknesses of this submission outweigh the strengths by a good margin. The novelty and empirical validation presented would seem to be more in line with a workshop submission than a full paper.
1. The suggested method is more trivial than not and I am somewhat skeptical that very similar and/or identical methods do not already exist in the literature. However, even if we take it for granted that nothing like this has been done before, the empirical comparisons are too shallow, see below.
2. The method isn't specified in sufficient detail. We are told (line 143ff) that "the parameters of both the NN and the kernel function are learned via gradient-based optimization methods." OK but with what objective function? Mean squared error for regression tasks? Something else? Does the objective function matter?
3. The number/variety of benchmark algorithms used in experiments is inadequate. What about a MLP that is given access to cyclical/fourier features in the time domain in the experiments in Sec. 5.3? What about various neural processes, e.g. reference [1] below? What about attention-based mechanisms? The omission of the latter is particularly troubling given the empirical success of attention-based methods and the similarity of attention-based mechanisms with your inner product construction. Given the simplicity of the proposed construction, a thorough empirical assessment (using a variety of datasets and benchmark methods) is absolutely crucial to convince the reader that the proposed method is useful in practice. In general the appendix reveals very little information about the MLPs used in the experiments. How many hidden units? How were these trained? etc. The proposed method may be interesting, but these toy experiments do not convince.

References
1. "Convolutional Conditional Neural Processes,"
Jonathan Gordon, Wessel P. Bruinsma, Andrew Y. K. Foong, James Requeima, Yann Dubois, Richard E. Turner
2. "Attention Is All You Need,"
Ashish Vaswani, Noam Shazeer, Niki Parmar, Jakob Uszkoreit, Llion Jones, Aidan N. Gomez, Lukasz Kaiser, Illia Polosukhin

---

> ### Author Response · Authors · 2022-08-02
> **Response to Reviewer qDxM, part 2 of 2**
>
> -- The novelty and empirical validation presented would seem to be more in line with a workshop submission than a full paper. The suggested method is more trivial than not and I am somewhat skeptical that very similar and/or identical methods do not already exist in the literature.
>
> We disagree on the lack of novelty of our work. As stated in our introduction, to the best of our knowledge, no existing approaches allows a modeler to choose any appropriate kernel of known prior information from multiple data sources. If the reviewer believes that an identical method exists in the literature, we would sincerely appreciate a reference. If not, we would appreciate the reviewer removing this comment.
> We would also argue that by no means simplicity equates to lack of novelty or unconvincingness. Our final framework is presented in a simple and elegant form, but it was derived from a novel model and required a complete theoretical derivation to motivate it, which does not seem trivial to us.  Furthermore, we would argue that the fact that our theoretical motivation yields a comparatively simple algorithm is a positive aspect of this algorithm.  The machine learning field frequently adapts relatively simple modifications over more complex contributions; for example, consider skip-connections (e.g., resnet), which are straightforward to implement but have been widely adopted.

---

> ### Author Response · Authors · 2022-08-02
> **Response to Reviewer qDxM, part 1 of 2**
>
> -- The method isn't specified in sufficient detail. We are told (line 143ff) that "the parameters of both the NN and the kernel function are learned via gradient-based optimization methods." OK but with what objective function? Mean squared error for regression tasks? Something else? Does the objective function matter?
>
> We use MSE objective for Section 5.1 and 5.2 and MAE objective for Section 5.3 (to put less weight on outliers and thus enhance the model performance). Please see Appendix G in the revised version of our paper for details of model structures and training procedure. We believe that the primary contribution to the predictive performance is our model formulation. We also note that code was included with the submission that documents all the detailed modeling choices and implementation.
>
> -- The number/variety of benchmark algorithms used in experiments is inadequate. What about a MLP that is given access to cyclical/fourier features in the time domain in the experiments in Sec. 5.3? What about various neural processes, e.g. reference [1] below? What about attention-based mechanisms?
>
> We find the reviewer’s observation that attention-based methods have mathematical similarities to our proposed approach quite intriguing, and we thank the reviewer for pointing it out.  While we note that we are unaware of any attention-based mechanisms that were explicitly designed to model similar data structures, we have attempted to construct attention-based models to address this criticism. In the revised version of our paper, we add 4 benchmarks that leverages the Vision Transformer (ViT) structure to process the remote sensing data in Section 5.2. However, ViTs are not directly applicable to data with multiple information sources/modalities. Therefore, we again combine them with Random Forests (RFs) to form a joint model, expand the temporal information, and again consider the sinusoidal trick that was used in other baselines. The results in the updated Table 1 show that ViT-based models have better performance than CNN-based models, but still underperform ICKy. Furthermore, we replaced the NN part in ICKy with ViT-based structure, which had minimal impact on performance.
> Additionally, in Section 5.3, we have added a cyclic MLP and two Gaussian Neural Processes (GNPs) benchmarks. It appears that all these benchmarks yield much larger prediction errors than ICKy. We believe these additional benchmark methods help demonstrate the strength of ICKy and thus make our claim more convincing.
>
> -- In general, the appendix reveals very little information about the MLPs used in the experiments. How many hidden units? How were these trained? etc.
>
> For synthetic data, the MLP consists of just one single hidden layer of width 1000. For remote sensing data, the last layer of the backbone NN for ICK and all benchmark models has 1000 hidden units. It is then followed by one or two hidden layers with 512 units (two hidden layers for benchmark models and one hidden layer for ICK). For UCI ML repository data, all MLPs contain 3 hidden layers of width 128, 32, 32, respectively. Please see Appendix G for the details of MLPs and training procedure (i.e. optimizer, loss function, etc.) in the revised version of our paper.
>
> -- For M>2 the authors suggest using a "chained inner product" to form predictions (Appendix A). The authors also state (line 295ff) that "A major limitation of ICK lies in our method of combining latent representations as the nature of inner product (i.e. the effect of multiplying small numbers) may cause vanishing gradient problems when we have a large number of sources of information (i.e. M is large)." Have you tried M>3? When do you expect ICK to break down?
>
> In Figure A1(c), we can see that the true and predicted values of the target are almost aligned with the line y = x. We tried M = 4 in our experiment but did not observe substantial improvement upon the results of M = 3. When M > 4, the gradient over both the parameters of NN and GP starts to change very slowly. Therefore, we do not recommend to set the value of M greater than 4.  There are some additional tricks that could be used to make this scale to a higher value of M, such as combining modalities with a single Nyström map, but we have not evaluated these schemes and this is left as future work.
>
> -- What happens when Assumption 1 (line 158ff) does not hold? Can you test this experimentally with an artificial dataset that breaks this assumption?
>
> Assumption 1 is a rather mild assumption (see response to Review 1) that holds true with most initialization procedures.  We have not determined an experiment that breaks this structure in a natural way, so it is difficult to test.

---

> > ### Comment · Reviewer_qDxM · 2022-08-09
> > **re: author response**
> >
> > i thank the authors for their response. i think it's valuable that the authors added some additional empirical evaluations. in my opinion, this is still less empirical evaluation than we would see in an ideal neurips submission of this kind, but it is a step in the right direction. for this reason i have raised my score. frankly, i'm surprised that at least two of the reviewers are apparently satisfied with the empirical evaluation included in the original submission.
> >
> > in particular, given the details provided in the draft i'm unable to determine if the attention-based models that were added make use of attention in a way that is canonical. my concern is that this may not be the case, but it is difficult to evaluate.
> >
> > the bayesian deep learning literature is gigantic, uses a mix of different languages and, consequently (and unfortunately) can be somewhat difficult to follow. i recall at least one work that was somewhat similar to the present work, but unfortunately i am unable to come up with a reference. for what it's worth, here are some additional references that may be of interest to the authors:
> >
> > - https://arxiv.org/abs/2011.12829
> > - https://dl.acm.org/doi/abs/10.5555/3546258.3546415

---

### Official Review · Reviewer_pXCj · 2022-07-11

**Rating:** 5
**Confidence:** 4
**Soundness:** 3 good
**Presentation:** 3 good
**Contribution:** 3 good

**Summary:**

The paper proposes to blend the strengths of deep learning and the clear modeling capabilities of GPs by using a composite kernel that combines a kernel implicitly defined by a neural network with a second kernel function chosen to model known properties. Technically, the authors approximate the resultant GP by combining a deep network and an efficient mapping based on the Nyström approximation. Results on both synthetic and real-world data sets show the superior performance and flexibility of the proposal.

**Questions:**

See Weaknesses

**Limitations:**

the authors have adequately addressed the limitations

**Strengths And Weaknesses:**

Strengths
- The paper proposes a practical method to incorporate prior knowledge into NNs. It offers an effective way for processing data with multiple sources of information where some explicit priors can be introduced in the form of GPs.
- The paper establishes a way to integrate NNs and GPs efficiently and train them jointly.
- The empirical results are interesting and insightful.


Weaknesses
- Assumption 1 seems to apply to NN-GP kernels but does not apply to the approximate NN-GP kernels corresponding to finite-width NNs.
It does not apply to typical kernels like RBF/polynomial ones as well. Can you clarify its validity theoretically/empirically?

- In the case of an NN (whose parameters are $\theta^{(1)}$) plus a classic kernel (whose parameters are $\theta^{(2)}$), the learning is to find a good distribution $p(\theta^{(1)})$ and a point-estimate $\theta^{(2)}$? If so, can you clarify how you parameterize $p(\theta^{(1)})$? (E.g., a variational BNN with Gaussian variational whose mean is fixed as 0?) If not, and if you just learning a point-estimate $\theta^{(1)}$, Lemma 1 would not apply and Theorem 2 does not hold.

- The loss function for training is just the MSE error? Can the framework be extended to handle classification problems? It seems that it is non-trivial as the output should only be 1-D.

- How to pre-define the integrating points? Can they be updated by the training signals?

- How does the proposal compare to the recent work NeuralEF?

NeuralEF: Deconstructing Kernels by Deep Neural Networks. ICML 2022.

---

> ### Author Response · Authors · 2022-08-02
> **Response to Reviewer pXCj, part 2 of 2**
>
> -- How to pre-define the integrating points? Can they be updated by the training signals?
>
> There are several approaches to pre-define the integrating points in the literature, such as uniform sampling without replacement [1], adaptive sampling [2], and pseudo-inputs [3]. In our experiments, we take evenly spaced points over the input space and use them as integrating points (which can be viewed as a simplified approach to generate pseudo-inputs) as described in Section 4.2.1. For example, if we believe the input has a periodic relationship with T = 100 with the output and we want to use 10 integrating points, then these points are simply defined as [0, 10, 20, …, 90]. We take this approach because it gives empirically better results compared to other sampling methods.
> In the current implementation, the integrating points are treated as hyperparameters and are not updated.  We note that there is prior work to suggest that the integrating or inducing points can be learned in a joint framework (e.g, [4]).  We are unaware of any reason why these learning approaches could not be adapted for our situation, but we have not empirically evaluated this strategy. As our performance has largely saturated with the current number of integrating points (see Supplemental Figure E1), we would not expect a large improvement by learned better points.
>
> Reference:
> [1] Kumar, Sanjiv, Mehryar Mohri, and Ameet Talwalkar. "Sampling techniques for the nystrom method." Artificial intelligence and statistics. PMLR, 2009
> [2] Deshpande, Amit, and Santosh Vempala. "Adaptive sampling and fast low-rank matrix approximation." Approximation, Randomization, and Combinatorial Optimization. Algorithms and Techniques. Springer, Berlin, Heidelberg, 2006. 292-303.
> [3] Snelson, Edward, and Zoubin Ghahramani. "Sparse Gaussian processes using pseudo-inputs." Advances in neural information processing systems 18 (2005).
> [4] Li, Yitong, et al. "Targeting EEG/LFP synchrony with neural nets." Advances in Neural Information Processing Systems 30 (2017).
>
> -- How does the proposal compare to the recent work NeuralEF?
>
> The NeuralEF paper proposes to estimate the kernel, including NNGP kernel and NTK, by using a series of NNs to approximate eigenfunctions under the principle of eigen-decomposition. However, in our proposal, instead of estimating the kernel itself, we use the Nystrom method to map the kernel matrix into a latent space with desired dimension. Therefore, we do believe that NeuralEF is highly related to our proposed approach. We have included it in the Related Work section, but have not implemented it as a benchmark model for comparison.

---

> ### Author Response · Authors · 2022-08-02
> **Response to Reviewer pXCj, part 1 of 2**
>
> -- Assumption 1 seems to apply to NN-GP kernels but does not apply to the approximate NN-GP kernels corresponding to finite-width NNs. It does not apply to typical kernels like RBF/polynomial ones as well. Can you clarify its validity theoretically/empirically?
>
> Assumption 1 does hold in the finite width neural network case under zero-mean iid random parameters, which is the typical initialization strategy.  To see this, note that z_i will be determined by the inner product of a random vector w_i^L and the output of the previous layer, where L is the number of layers in the neural network, and z_j is likewise determined by w_j^L.  If w_i^L and w_j^L are zero-mean random independent variables, then the expectation of the products turns into the product of the expecations, then the covariance of z_i and z_j will be 0, as E[w_j^L]=E[w_i^L]=0.  Likewise, this assumption is straightforward for random Fourier features, as each one of the random Fourier features integrates to 0 with the other features in expectation.  This is more challenging to prove for the Nyström approximation but seems to reasonably hold in practice.  We do note that Assumption 1 only holds over the expectation of the a priori parameter distribution, and we do not claim it holds for realized values nor after training, as discussed in our limitations.
>
> -- In the case of an NN (whose parameters are θ(1)) plus a classic kernel (whose parameters are θ(2)), the learning is to find a good distribution p(θ(1)) and a point-estimate θ(2)? If so, can you clarify how you parameterize p(θ(1))? (E.g., a variational BNN with Gaussian variational whose mean is fixed as 0?) If not, and if you just learning a point-estimate θ(1), Lemma 1 would not apply and Theorem 2 does not hold.
>
> In our algorithm, we initialize parameters from a zero-mean scaled random Gaussian and proceed with learning a point estimate.  We did implement this approach using Variational Bayesian layers to estimate a Gaussian posterior over the parameters, but because we did not see an increase in performance we preferred the simpler point-estimate formulation.  Regarding Lemma 1, we state that this property holds a priori, but as we discuss in our limitations, we do not claim that it holds after the network is learned.  Rather, we use this as motivation to choose our model form.  We note that existing empirical results suggest that training sufficiently wide NNs yields similar models to their infinite GP equivalent [1], so we believe that this is a reasonable strategy for model construction, and our empirical results support this conclusion.  Future work will consider Bayesian learning strategies in more detail.
>
> Reference: [1] Lee, Jaehoon, et al. "Deep neural networks as gaussian processes." ICLR (2017).
>
> -- The loss function for training is just the MSE error?
>
> We use MSE objective for Section 5.1 and 5.2 and MAE objective for Section 5.3, which puts less weight on outliers and thus enhance the model performance. Please see Appendix G for all model architectures and training details in the revised version of our paper.
>
> -- Can the framework be extended to handle classification problems? It seems that it is non-trivial as the output should only be 1-D.
>
> The framework can be extended to handle classification problems based on existing approaches for multi-class classification from GPs. Briefly, a separate ICK architecture can be run for each class, and then the scalar output for each class can be concatenated and put through a softmax to create class probabilities. Please see Appendix H for more details and a proof-of-concept experiment on Rotating MNIST on how our method can be adapted for classification.

---

> > ### Comment · Reviewer_pXCj · 2022-08-08
> > **Further question**
> >
> > It is glad to see your reply. I am largely satisfied with it.
> >
> > Yet, as you agreed, Assumption 1 holds only in the prior case and would fail in the course of training. And you only learn a point-estimate $f(, theta1)$. Do these mean the learning outcome is not a composite Kernel GP anymore? If so, it is better the clarify that Theorem 1 serves as a motivation for this work instead of a theoretical basis, and you should revise arguments like "Then, we approximate the resultant GP by combining a deep network and an efficient mapping based on the Nyström approximation". In my opinion, during training, you don't approximate the composite Kernel GP as Theorem 1 doesn't hold, right?

---

> > > ### Author Response · Authors · 2022-08-08
> > > **Reply to question from Reviewer pXCj**
> > >
> > > Thank you so much for your question!
> > >
> > > To clarify, we do not expect Theorem 1 to hold exactly after training, but we would expect that the learned network is much closer to the composite GP due to the model structure.  We tried to be clear with the limitations in our writing. We only claimed that the properties held a priori on lines 146-148 by stating "we first analytically show that our ICK framework is approximately equivalent to a composite GPR model a priori using a multiplicative kernel between the kernel implicitly defined by the NN on $x^{(1)}$ and the chosen kernel on $x^{(2)}$" and explicitly stated that we did not expect it to hold a posteriori in the limitations. Regardless, to prevent any misconceptions on the theoretical claims, we have added the following clarifying text on lines 148-150: "This theory is used to motivate the model form.  The model will deviate from the GP solution after learning, but we note that recent work suggests that the predictions from a trained neural network may not vary too much from its GP equivalent [1]."
> > >
> > > Reference:
> > > [1]. Lee, Jaehoon, et al. "Deep neural networks as gaussian processes." arXiv preprint arXiv:1711.00165 (2017).

---

> > > > ### Comment · Reviewer_pXCj · 2022-08-09
> > > > **Response**
> > > >
> > > > Thanks for the reply. I disagree that NNGP [1] sheds light on this work. The difference is obvious. In NNGP, the Gaussian prior on weight space corresponds to a GP in function space, and then the training of *Bayesian neural networks* amounts to the exact analytical Bayesian inference in the GP form. But this work, in practice, learns a point-estimate model, thus it is meaningless to say the model approaches the NNGP posterior after training.
> > > >
> > > > It is possible to tell the story from the "sample-then-optimize" [*1, *2] viewpoint where the point estimate is regarded as a posterior sample corresponding to some GP prior. It will make the whole paper more self-contained.
> > > >
> > > >
> > > > [*1] Bayesian Deep Ensembles via the Neural Tangent Kernel. Bobby He et al.
> > > > [*2] Bayesian Inference with Anchored Ensembles of Neural Networks, and Application to Exploration in Reinforcement Learning
> > > > Tim Pearce et al.

---

> > > > > ### Author Response · Authors · 2022-08-09
> > > > > **Reply to response**
> > > > >
> > > > > Thank you for your suggestion!  The sample-then-optimize framework is more closely to our proposed model and would make our work more self-contained. We will update the references and text accordingly. For clarity, we note that the NNGP reference compares a GP to an SGD point estimate, not a BNN. They state this conclusion on their page 7, "We additionally find the performance of the best finite-width NNs, trained with a variant of SGD, approaches that of the NNGP with increasing layer width..." It hints at a possible relationship between SGD and Bayesian inference in certain regimes – were the neural networks trained in a fully Bayesian fashion, rather than by SGD, the approach to NNGP in the large width limit would be guaranteed.

---

### Author Response · Authors · 2022-08-02
**Response to all reviewers**

We thank all the reviewers for their insightful comments. We have listed our answers to your questions below.  We have also uploaded a revised manuscript that addresses many of the questions and comments from all reviewers, including several clarifications and many additional competing methods. Major additions and edits are marked in blue, and we highlight below when we made specific manuscript edits. The anonymous code repository has also been updated to reflect the new comparison methods.

---

### Meta-Review · Area_Chair_SthP · 2022-08-29

**Recommendation:** Reject
**Confidence:** Certain

**Metareview:**

The submission considers fusing prior knowledge into neural networks by *modulating* the learnt features. The modulation is either additive or multiplicative using another set of features outputted by a kernel-based mapping with linear/periodic kernels. The method is akin to using composite kernels (or combining kernels) in the GP literature and is tested on several regression benchmarks.

The reviewers acknowledged the method is simple and seems to give a desirable performance when the prior knowledge is known. I (AC) read the paper and have several concerns (some of which have already been raised by one or more reviewers):
- the clarity around the connection to composite GP kernels could be improved (there was also a question about the goal of this connection and the authors stated that it is to motivate the proposed approach),
- many desirable properties of GPs (hyperparameter learning using the marginal likelihood, predictive uncertainty) seem to be lost since the objective function is only MSE/MAE and the parameters are not treated probabilistically [the authors promised to consider this in the next iteration].
- limited experiments to show the capability of the ICK framework beyond two data sources, and for various forms of prior knowledge beyond seasonality and the PM 2.5 forecasting task. Some further analyses of the forecasting results + errorbars would be appreciated.

Despite the simplicity of the approach, for the above reasons, I feel that the submission is not ready for publication. I hope, however, to see an updated version published at a conference soon. The reviews are already fairly positive and the discussions here could be useful for polishing up the experiments and writing.


**Award:**

No

---

### Decision · Program_Chairs · 2022-09-14

Reject